# Cervical Secretion Methylation Is Associated with the Pregnancy Outcome of Frozen-Thawed Embryo Transfer

**DOI:** 10.3390/ijms24021726

**Published:** 2023-01-15

**Authors:** Yi-Xuan Lee, Po-Hsuan Su, Anh Q. Do, Chii-Ruei Tzeng, Yu-Ming Hu, Chi-Huang Chen, Chien-Wen Chen, Chi-Chun Liao, Lin-Yu Chen, Yu-Chun Weng, Hui-Chen Wang, Hung-Cheng Lai

**Affiliations:** 1Graduate Institute of Clinical Medicine, College of Medicine, Taipei Medical University, Taipei 11030, Taiwan; 2Taipei Fertility Center, Taipei 11030, Taiwan; 3Translational Epigenetics Center, Shuang Ho Hospital, Taipei Medical University, New Taipei 23504, Taiwan; 4Department of Obstetrics and Gynecology, Shuang Ho Hospital, Taipei Medical University, New Taipei 23504, Taiwan; 5International Ph.D. Program for Cell Therapy and Regeneration Medicine, College of Medicine, Taipei Medical University, Taipei 110301, Taiwan; 6Department of Obstetrics and Gynecology, Hai Phong University of Medicine and Pharmacy, Hai Phong 04254, Vietnam; 7Department of Obstetrics and Gynecology, School of Medicine, College of Medicine, Taipei Medical University, Taipei 11303, Taiwan; 8Division of Reproductive Medicine, Department of Obstetrics and Gynecology, Taipei Medical University Hospital, Taipei 11030, Taiwan; 9Department of Obstetrics and Gynecology, Tri-Service General Hospital, National Defense Medical Center, Taipei 11490, Taiwan

**Keywords:** cervical secretion, methylation, implantation, IVF-FET, non-invasive

## Abstract

The causes of implantation failure remain a black box in reproductive medicine. The exact mechanism behind the regulation of endometrial receptivity is still unknown. Epigenetic modifications influence gene expression patterns and may alter the receptivity of human endometrium. Cervical secretions contain endometrial genetic material, which can be used as an indicator of the endometrial condition. This study evaluates the association between the cervical secretion gene methylation profile and pregnancy outcome in a frozen-thawed embryonic transfer (FET) cycle. Cervical secretions were collected from women who entered the FET cycle with a blastocyst transfer (36 pregnant and 36 non-pregnant women). The DNA methylation profiles of six candidate genes selected from the literature review were measured by quantitative methylation-specific PCR (qMSP). Bioinformatic analysis of six selected candidate genes showed significant differences in DNA methylation between receptive and pre-receptive endometrium. All candidate genes showed different degrees of correlation with the pregnancy outcomes in the logistic regression model. A machine learning approach showed that the combination of candidate genes’ DNA methylation profiles could differentiate pregnant from non-pregnant samples with an accuracy as high as 86.67% and an AUC of 0.81. This study demonstrated the association between cervical secretion methylation profiles and pregnancy outcomes in an FET cycle and provides a basis for potential clinical application as a non-invasive method for implantation prediction.

## 1. Introduction

Ever since the first baby was born via in vitro fertilization (IVF) in 1987, this assisted reproduction technology has become the most effective method for couples with difficulty conceiving, and the number of IVF treatments each year is rising rapidly worldwide [1]. Despite the significant improvements in IVF technology with respect to embryo selection, such as euploid embryo selection with preimplantation genetic testing for aneuploidy (PGT-A), successful pregnancy is not guaranteed, even when euploid blastocysts are transferred. Studies have shown that euploid embryos resulted in only 50% ongoing pregnancies per transfer and a 77% cumulative live birth rate after multiple transfer attempts [2,3]. Factors other than those related to embryos must be considered; consequently, implantation outcomes may be improved after correcting such variables.

The initiation of pregnancy begins with the successful implantation of an embryo, which, in turn, is initiated by the synchronized crosstalk between a well-developed blastocyst and a receptive endometrium [4]. The ability of the endometrium to allow for embryonic implantation is termed endometrial receptivity. Implantation failure related to endometrial receptivity remains a proverbial black box in reproductive medicine, and the exact regulatory mechanism of endometrial receptivity is still unknown. This frustrating phenomenon for both patients and clinicians has remained poorly characterized and is considered a major cause of infertility [5]. Studies have suggested that approximately two-thirds of implantation failures are due to endometrial receptivity defects, while the embryo quality itself is responsible for the remaining one-third [6,7].

Normal implantation involves a complex sequence of signaling events and only occurs during a specific time-span in the mid-secretory phase, termed the “window of implantation (WOI)”, which represents the period of maximum uterine receptivity for implantation [8]. The WOI has inter-individual variation but typically starts on day 19 or 20 of the menstrual cycle, or on the 7th day after the luteinizing hormone (LH) surge (LH + 7), and lasts for the next 4–5 days [9]. Omics technologies, such as transcriptomic approaches, have been used to determine the transcriptomic signature of the endometrium during the WOI [10]. However, only 18.6% of recurrent implantation failure (RIF) patients experience a displaced (asynchronous) WOI alone, and its influence on implantation remains doubtful [11]. Thus, a better method for predicting endometrial receptivity in IVF cycles is needed.

Epigenetic regulation plays a role in regulating gene expression in the human endometrium during the menstrual cycle, and aberrant epigenetic patterns may be associated with implantation failure and other endometrial pathologies such as endometrial cancer [12,13,14]. In contrast to the genetic regulation of gene function via the alteration of a DNA sequence, epigenetic regulation is defined as stably heritable changes in chromosomes without alterations in the DNA sequence [15]. Environmental factors such as nutrition, exercise, substances in the environment, and exposure to chemicals can influence the establishment and maintenance of epigenetic patterning [16]. Increasing evidence also hints at the important roles of epigenetic regulation, such as DNA methylation, in endometrial receptivity and the process of embryonic implantation [17,18,19]. DNA methylation is an epigenetic modification that is achieved by the addition of a methyl group (-CH3) to the fifth carbon of the cytosine ring [20,21]. DNA methylation mainly occurs in CpG sites and has been characterized as a crucial mediator of fundamental biological functions, such as embryonic development, carcinogenesis, aging, and endometrial regeneration [22,23].

DNA methylation is initiated at the beginning of endometrial development and remains almost unchanged throughout the menstrual cycle until the late-secretory phase and in menstruation during the shedding of the old endometrium [23]. Endometrial stem cells are responsible for cyclic regeneration, remodeling, and degradation following shedding and menstruation in each cycle [24,25]. From the beginning of menstruation, the reconstruction of a new endometrium is accomplished by a resident stem cell colony under vigorous epigenetic reprogramming [23,26]. Through these factors, DNA methylation changes intensely only when endometrial stem cells start to regenerate a new endometrium. Each newly grown endometrium may have a distinct DNA methylation pattern to regulate its biological behavior, including its receptivity for embryonic implantation. The consistency of DNA methylation during the menstrual cycle makes it possible to predict endometrial receptivity at the beginning of the secretory phase.

Endometrial tissue obtained through invasive biopsies have been required in most studies investigating endometrial receptivity. Nevertheless, the inter-cyclical variation in the condition of the endometrium is being ignored arbitrarily, as evidenced by the inconsistent results obtained from different menstruation cycles in the same individuals [27]. To investigate endometrial receptivity in the same cycle, a non-invasive approach is necessary. Cervical secretions have also been observed to contain implantation-related cytokines and various growth factors that are produced by a receptive endometrium [28,29,30] and may be used as a non-invasive indicator reflecting the condition of the endometrium. Indeed, we have used the DNA methylome in cervical secretions as a non-invasive biomarker to detect endometrial cancer [31,32]. This evidence has demonstrated the potential of using cervical secretions as a non-invasive proxy for the investigation of the DNA methylation profile and endometrial receptivity in the same cycle.

Collectively, we hypothesized that the DNA methylation profiles are different between receptive and non-receptive endometria confirmed by an individual’s pregnancy status and can be detected in cervical secretions. In this study, we aimed to investigate the DNA methylation profiles of selected candidate genes in cervical secretions and to correlate them with pregnancy outcomes.

## 2. Results

### 2.1. Candidate Gene Selection

We chose the candidate genes via a literature review and search using the keywords “endometrial receptivity,” “implantation,” and “DNA methylation.” Only human studies were incorporated. Several studies reported significant roles of *HOXA10* in implantation and endometrial receptivity [33,34,35,36]. *HAND2* [37,38,39] has been reported to play roles in the peri-implantation endometrium. Kukushkina et al. [40] reported significant methylation differences in *KSR1*, *PPT2*, *PRKAG2*, and *ZMIZ1* between the pre-receptive and receptive endometria. Therefore, we selected these six candidate genes in out pilot study.

### 2.2. CpG-Level Differential Methylation Analysis of Candidate Genes between Receptive Endometrium Compared to the Pre-Receptive Endometrium

To determine the impact of DNA methylation on these six genes with respect to endometrium receptivity, we analyzed the DNA methylation statuses reported in the previously published data [40]. These data concerned the investigation of DNA methylation in the endometria from biopsies performed on 17 healthy, fertile-aged women between the pre-receptive and receptive phase in the same menstruation cycle using genome-wide technology (NCBI public database, GSE90060). We used the β values to represent the DNA methylation levels of each CpG site in the candidate genes and tested the differences between pre-receptive and receptive endometria. The result showed that the β-value was statistically different in several CpG sites of the six candidate genes (Figure 1). These data suggest that the methylation of these genes may be associated with endometrial receptivity.

### 2.3. DNA Methylation Analysis of Candiadate Genes via Methylation Array

In our previous work, we found that the genome-wide methylation profiles in cervical secretions differ between pregnancy and non-pregnancy cycles [41]. Since endometrial receptivity constitutes the ability of the endometrium to allow blastocysts to successfully implant themselves and to be nourished and sustained, we assumed that the methylation pattern of candidate genes might be associated with the pregnancy outcome in the FET cycle. To this end, we collected cervical secretion DNA methylome data from 41 women receiving FET [41]. Of the 41 women, 24 were successful in pregnancy, and 17 were not. The patients’ characteristics are described in Table 1. The other parameters that may affect pregnancy outcomes, including age, the etiology of infertility, endometrial thickness on the day of the FET, the number of embryo transfers, and the FET regimen, all showed no statistically significant differences between the two groups.

The association of pregnancy outcome and candidate gene methylation was evaluated by analyzing the area underneath the receiver operating characteristic (ROC) curve (AUC) (Appendix A). The top 10 CpG probes in each candidate gene revealed that the AUCs ranged between 0.6–0.84 (Figure 2). These data indicated the potential for DNA methylation in candidate genes to be used to identify the degree of endometrial receptivity between the pregnant and non-pregnant groups. This proof-of-concept approach demonstrated that pregnancy outcomes in an FET cycle were associated with changes in DNA methylation profiles and could be detected in cervical secretions, thus constituting a non-invasive approach.

### 2.4. Candidate Genes’ DNA Methylation Analysis via Quantitative Methylation-Specific Polymerase Chain Reaction (qMSP)

Next, we measured the methylation levels of the six candidate genes with eight regions via the qMSP platform in the cervical secretions from 72 samples. A total of 36 pregnant and 36 non-pregnant women were enrolled in this case-control study. Their characteristics are described in Table 2. There was no significant difference between the two groups regarding basic characteristics, including age, the etiology of infertility, endometrial thickness on the day of the FET, the number of embryonic transfers, and the FET regimen.

Among the eight regions, KSR1-MS02 showed significant hypermethylation in the non-pregnancy group compared with the pregnancy group (Figure 3). Other genes also showed varying degrees of differential methylation statuses between the pregnancy and non-pregnancy groups but without statistical significance (Figure 3). The AUCs for a single gene/region ranged from 0.55–0.67 (Figure 4), indicating the possible value of categorization between the pregnancy and non-pregnancy groups.

### 2.5. Machine Learning Approach for Differential Methylation Analysis of Candidate Genes

Machine learning is a valuable method that can be used to mine information hidden in high-dimensional omics datasets, and it has been facilitated by the rapidly increasing computing power employed to deal with high-throughput omics data [42]. The use of machine learning can compensate for the insufficiency of conventional statistics modelling and has become famous for data analysis, interpretation, and the identification of biomarkers for diagnostic or predictive tasks [43]. We used a supervised machine learning approach with different algorithms and combined the six candidate genes with eight regions. The results showed that the AUC values ranged from 0.57–0.89 with various machine learning models (Figure 5). Among them, the logistic regression model ranked highest with an accuracy of 86.67% and an AUC of 0.81, followed by the multilayer perceptron model with an accuracy of 73.33% and an AUC of 0.89 (Figure 5, Table 3).

## 3. Discussion

This study demonstrates that the pregnancy outcomes in an FET cycle are associated with changes in DNA methylation profiles precipitated by non-invasive cervical secretions. Using the machine learning approach, the combination of the candidate genes’ DNA methylation profiles yields an accuracy as high as 86.67% for the categorization of pregnancy status. This promising result may contribute to the possibility of predicting a successful pregnancy using DNA methylation biomarkers.

The endometrial methylome has been shown to be dynamic and change within and between different menstruation cycles [13,23], and may be influenced by the supraphysiologic hormone milieu during IVF/ET treatment [44,45]. However, whether these methylation differences derive from either true physiological changes or individual variability remains elusive. DNA methylation changes occur during the transition from the pre-receptive to the receptive phase in the human endometrium [40]. These dynamic methylation changes involve endometrial functions, including implantation. Abnormal methylation changes may result in endometrial dysfunction and implantation failure. In our results, we demonstrated that the DNA methylation profile is associated with endometrial receptivity, as evidenced by the differential methylation changes with pregnancy status detected in cervical secretions.

Several endometrial receptivity prediction tools have been developed and commercialized for the personalized timing of embryonic transfer according to the time of the WOI shift [10,46,47,48]. These tools use the transcriptomic signature profile to suggest the time for embryonic transfer. Although the timing for embryonic transfer is essential, it is not a limited time interval [49]. Recently, data from two large trials and a meta-analysis showed no clinical benefits associated with endometrial receptivity analysis of the WOI shift [50,51,52]. The nature of the WOI shift in different menstruation cycles may be governed by normal inter-cycle variation instead of pathological changes [27]. The endometrium can compensate for the shifted WOI and transform for implantation [53]. This evidence suggests that the timing of embryonic transfer may not be the critical factor of successful implantation; the situation of the endometrium may have a greater impact.

The endometrium undergoes cycles of growth and regression in each menstrual cycle, and adult progenitor stem cells are likely responsible for this regenerative process [24,25]. The epigenetic regulation involved in adult stem cell division and these changes is inherited by all daughter cells, whereas other markers that arise during amplification or mature cells are lost when the functional layer is shed during menstruation [26,54]. Therefore, epigenetic reprogramming may occur in each menstrual cycle and impact endometrial receptivity at the beginning of the cycle. Our study focused on the methylation profile with respect to endometrial receptivity between pregnancy and non-pregnancy cycles. The identification of receptive endometria from each cycle may be more beneficial in terms of improving pregnancy outcomes in infertile patients than the timeframe of a WOI shift.

Most studies investigating endometrial receptivity require the use of invasive endometrial biopsies to obtain endometrial tissue. However, inter-cyclical variation cannot be evaluated through an invasive approach since endometrial biopsies preclude the possibility of an embryonic transfer occurring in the same cycle. This may cause inconsistent results and inconclusive benefits with respect to improving pregnancy outcomes. The analysis of cervical secretion was thought to be a non-invasive approach to determining the situation of the endometrium [31,32]. In contrast to invasive analysis, using cervical secretions avoids injuring the endometrium and implantation environment and allows for the investigation of endometrial receptivity in same conception cycle.

Taken together, this non-invasive analysis can not only be applied to both fresh or frozen embryonic transfer cycles but also to in vivo conception cycles in order to identify receptive endometria with better fecundity. Recently, the endometrial transcriptomic transformation across the whole menstruation cycle has been revealed at the single-cell level and has provided a more precise identification of functional genes’ regulation during WOI [55].Accordingly, we also investigated the differential methylation in selected WOI-related genes and its correlation with pregnancy outcomes (manuscript submitted). Combining all these data may unveil the intricate nature of implantation and the regulation of endometrial receptivity.

The candidate genes selected for this study are involved in various pathways in the implantation processes. *PRKAG2* (protein kinase AMP-activated non-catalytic subunit gamma 2) encodes the gamma-2 subunit of AMP-activated protein kinase (AMPK). This enzyme assists in sensing and responding to energy demands within cells. Previous studies have reported that *PRKAG2* may be affected by sex steroids and may be involved in the implantation process [56,57]. *HAND2* (heart and neural crest derivative-expressed protein 2) has been shown to be regulated by progesterone and is essential for implantation and decidualization [38,39]. Very little has been studied with respect to the functional roles of other candidate genes. Further research on these genes or protein products with regard to endometrial receptivity should be evaluated in the future.

The limitations of our study include its retrospective design and relatively small cohort size. The transferred embryos were not proven to be euploid, which might have affected the pregnancy outcomes. In addition, we obtained a cervical secretion sample on the day of embryonic transfer, that is, the period within the WOI that primarily reflects endometrial receptivity. For possible clinical applications, the assessment of endometrial receptivity should be conducted before embryonic transfer. Methylomic studies have shown that the DNA methylation status in endometrial tissue did not change significantly between the late-proliferative to mid-secretory phases [23,40]. Therefore, the earlier timing of the cervical secretion methylation pattern (late-proliferative or early secretory phase) may also be associated with endometrial receptivity. Further investigations regarding the earlier timing of the collection of cervical secretions are suggested for future studies.

## 4. Material and Methods

### 4.1. Ethics Statement

This study was reviewed and approved by the Taipei Medical University Joint Institutional Review Board (TMU-JIRB, approval number: N201703072). The samples were collected between August 2017 and December 2020. Written informed consent was obtained from all participating subjects and with the approval of the ethical committee. All research was performed in accordance with relevant guidelines and regulations.

### 4.2. Patients and Samples

Patients who entered the frozen embryonic transfer (FET) cycle with at least two high-quality embryos available were enrolled in this study. Blastocysts assessed by Gardner scoring system [58] with grade 2BB or above were defined as high-quality. Patients received FET preparation either with artificial cycle, also referred to as hormone replacement treatment (HRT) cycle, or nature cycle (NC) according to clinicians’ judgement. For HRT cycle, patients received oral administration of 6–8 mg of estradiol valerate daily for 8–12 days following menstruation days 2–3 until the endometrial thickness reached 7–14 mm and the serum estrogen level was within 200–400 pg/mL, thus mimicking natural conditions. Once the proliferation of the endometrium was considered sufficient, progesterone was administered to promote endometrial transition to secretory phase and prepare embryonic transfer according to embryos’ developmental stage. For the NC cycle, serum hormone and ultrasound monitoring were performed without medication prior to ovulation, and the transfer was scheduled to occur when the endometrium was synchronized to the embryos’ stage.

Cervical secretions were collected using a cotton ball on the day before embryonic transfer. Samples collected from 24 pregnant and 17 non-pregnant women were used in the pilot methylation array. Samples from 36 pregnant and 36 non-pregnant women were used for qMSP matched analysis. Successful pregnancy was defined as the presence of viable intrauterine pregnancy with at least one positive fetal heart beat lasting 10 weeks after embryonic transfer (equal to 12th weeks of gestation).

### 4.3. DNA Extraction, Bisulphite Conversion, and DNA Methylation Measurement

Cervical secretions were collected using a cotton ball placed in a 50mL centrifuge tube with an inner adaptor and stored at 4 °C. The cotton ball was rinsed with 1 mL phosphate-buffered saline (PBS) and then centrifuged at 1000× *g* for 10 min to collect the elution. Genomic DNA was extracted from cervical secretions using a QIAmp DNA Mini Kit (QIAGEN, Hilden, Germany). Bisulphite-converted DNA was treated using the EZ DNA Methylation Kit (Zymo Research Corp., Irvine, CA, USA) according to the manufacturer’s recommendations. PCR products were amplified using LightCycler 480 SYBR Green I Master (Roche, Basel, Switzerland) and LightCycler 480. The 20 μL reaction volume contained 2 µL of bisulphite-converted DNA, primers (250 nmol/L each), and 10 µL of the Master Mix. All specimens were tested in duplicate for each gene. The number of non-CpG regions of *COL2A1* in each independent methylation assay were used to normalize the input DNA. Then, the methylation level was calculated as follows [59]:dCp = (Cp of Gene) − (Cp of *COL2A1*) 

A smaller dCp indicates a higher methylation level. Test results with *COL2A1* Cp values > 36 were defined as detection failures. The primers were designed using Oligo 7.0 Primer Analysis software (Molecular Biology Insights, Inc., Colorado Springs, CO, USA). Primers used are shown in Appendix A.

### 4.4. Statistical Analyses and Machine Learning

Group analysis of continuous data, including patients’ characteristics and methylation levels, was conducted using the non-parametric Mann–Whitney U test. Correlations between categorical clinical variables and methylation levels were determined using Fisher’s exact test. Logistic regression was used to estimate the area under the receiver operating characteristic (ROC) curve associated with the methylation status of each candidate gene and gene combination. Statistical significance was set at a two-tailed *p*-value *<* 0.05. The above analyses and plots were performed and created using the statistical package in R (version 3.6.3) and SPSS (ver. 25, IBM Statistics, Armonk, NY, USA).

We used the R package Tidymodels to test the performance of the combination of candidate genes’ methylation in ten machine learning models. The ten models used were as follows: Logistic Regression, Naive Bayes, Support Vector Machines, K-Nearest Neighbor, Decision Trees, Random Forest, Bootstrap-Aggregating Trees, Light GBM, XGBoost, and Multilayer Perceptron. Predictive performance metrics, including AUC, accuracy, F_measure, precision, and recall, were calculated for the best config of each model with ten-fold cross-validation.

## 5. Conclusions

This study demonstrated that the DNA methylation profiles of cervical secretion are associated with pregnancy outcomes in an FET cycle. Our preliminary results showed that the combination of the candidate genes’ DNA methylation profiles could differentiate pregnancy from non-pregnancy samples with an accuracy as high as 86.67% through a machine learning approach. Further large-scale studies incorporating an earlier timing of the collection of cervical secretions for methylation profile analysis will be needed to confirm our results and their potential clinical use.

## Figures and Tables

**Figure 1 ijms-24-01726-f001:**
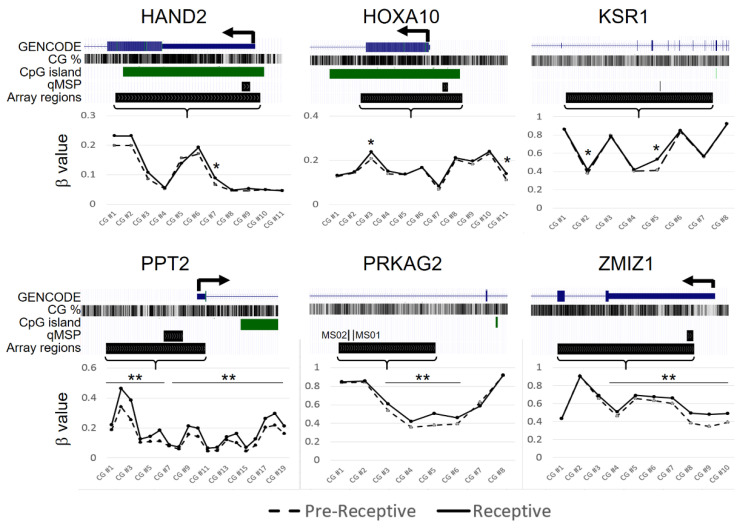
Bioinformatic analysis of six candidate genes (including seven CpG sites); the original genome-wide DNA methylation data originated from NCBI GEO Datasets (GSE90060). The dark-line represents receptive (LH + 7, within implantation window) endometrium, and the dashed line represents pre-receptive (LH + 2, early-secretory) endometrium. The CpG-level methylation analysis showed that the β-value was statistically higher in several CpG sites of the six candidate genes in receptive endometrium compared with pre-receptive endometrium. The β-values, qMSP amplified regions, and relative genomics features present locations including chr4:173528818-173530279 for HAND2, chr7:27173269-27174813 for HOXA10, chr17:27552177-27604383 for KSR1, chr6:32152427-32153135 for PPT2, chr7:151755057-151772238 for PRKAG2, and chr10:79310933-79315784 for ZMIZ1 according to the hg38 version. qMSP, quantitative methylation-specific PCR; chr, chromosome. * *p <* 0.05; ** *p <* 0.01.

**Figure 2 ijms-24-01726-f002:**
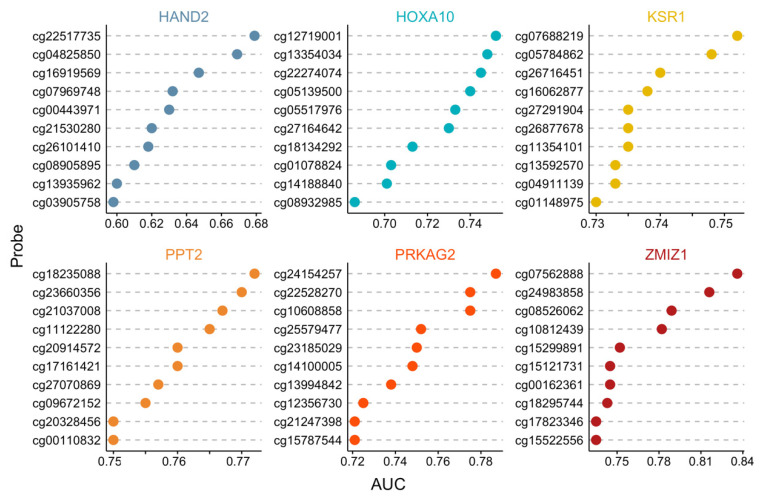
The AUC of top 10 probes of differentially methylated CpG sites in cervical secretion samples from 24 pregnant and 17 non-pregnant subjects with MethylationEPIC BeadChip array.

**Figure 3 ijms-24-01726-f003:**
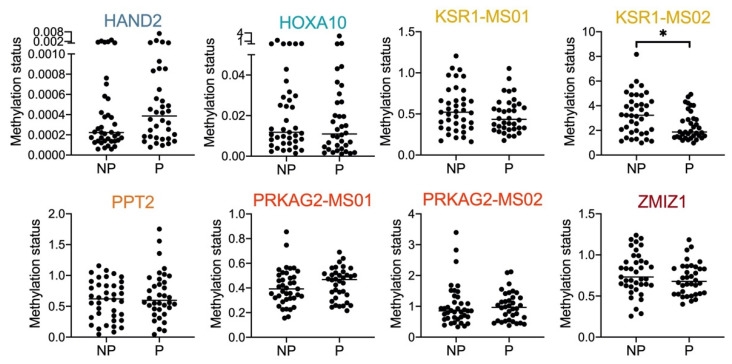
The methylation status in cervical secretion samples from 36 pregnant and 36 non-pregnant subjects with qMSP. * *p <* 0.05.

**Figure 4 ijms-24-01726-f004:**
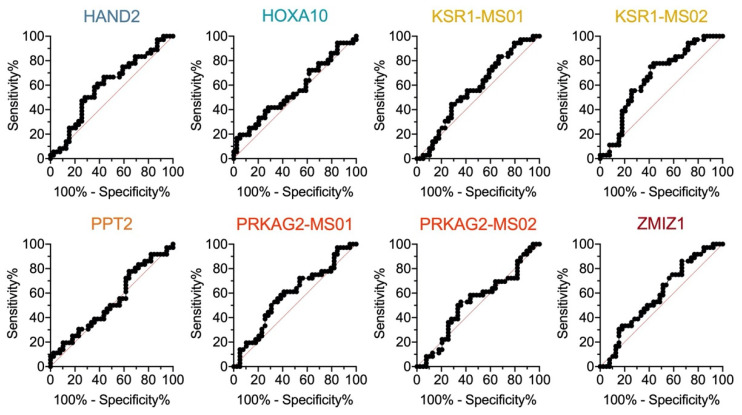
The AUC of single candidate gene/CpG site with respect to pregnancy prediction.

**Figure 5 ijms-24-01726-f005:**
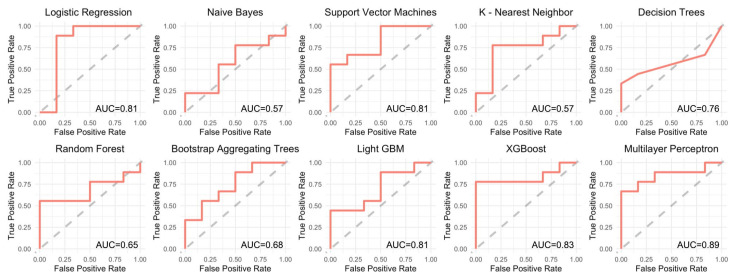
The combination of six candidate genes/8 CpG sites with different machine learning models. AUC values listed in right lower corner ranged from 0.57~0.89.

**Table 1 ijms-24-01726-t001:** Demographic characteristics of all clinical samples with methylation array.

	Pregnant (P)	Non-Pregnant (nP)	*p*-Value
Subject, n	24	17	
Age, years (range)	35.81 ± 1.87 (32–39)	35.97 ± 1.85 (32–38.8)	0.558
Infertility cause			0.868
Advanced Maternal Age	2 (8.33)	3 (21.43)	
Ovulatory	6 (25)	5 (29.41)	
Male factor	6 (25)	4 (23.53)	
Tubal factor	2 (8.33)	0 (0)	
Polycystic ovary syndrome	0 (0)	1 (5.88)	
Uterine	1 (4.17)	1 (5.88)	
Unexplained	5 (20.83)	2 (11.76)	
Endometriosis	2 (8.33)	1 (5.88)	
EM thickness (mm) during ET day	9.57 ± 2.62	9.24 ±2.36	0.659
Embryonic transfer number	2.0 ± 0.44	1.87 ± 0.64	0.28
Frozen embryonic transfer regimen			0.748
Hormone replacement therapy	9 (66.7)	8 (75)	
Nature cycle	15 (33.33)	9 (25)	

Data are expressed as mean ± standard deviation or n (%). *p* was calculated using the Mann–Whitney test for continuous data and Fisher’s exact test for count data. *p <* 0.05 denotes statistical significance (both two-sided).

**Table 2 ijms-24-01726-t002:** Demographic characteristics of all clinical samples determined via qMSP.

	Pregnant (P)	Non-Pregnant (nP)	*p*-Value
Subject, n	36	36	
Age, years (range)	37.93 ± 4.1 (31–46)	37.57 ± 4.0 (29.6–43.6)	0.558
Infertility cause			0.791
Advanced Maternal Age	12 (33.33)	7 (19.44)	
Ovulatory	4 (11.11)	7 (19.44)	
Male factor	4 (11.11)	6 (16.67)	
Tubal factor	0 (0)	1 (2.78)	
Polycystic ovary syndrome	2 (5.56)	2 (5.56)	
Recurrent pregnancy loss	0 (0)	1 (2.78)	
Unexplained	4 (11.11)	4 (11.11)	
Endometriosis	2 (5.56)	1 (2.78)	
Multiple	8 (22.22)	7 (19.44)	
EM thickness (mm) during ET day	9.48 ± 2.78	9.98 ± 2.2	0.659
Embryonic transfer number	2.0± 0.73	2.12 ± 0.83	0.28
Frozen embryonic transfer regimen			0.296
Hormone replacement therapy	24 (66.7)	27 (75)	
Nature cycle	12 (33.33)	9 (25)	

Data are expressed as mean ± standard deviation or n (%). *p* was calculated using the Mann–Whitney test for continuous data and Fisher’s exact test for countable data. *p <* 0.05 denotes statistical significance (both two-sided).

**Table 3 ijms-24-01726-t003:** The performance metrics of machine learning models.

	AUC	Accuracy	F_Measure	Precision	Recall
Logistic Regression	0.81	0.86667	0.88889	0.88889	0.88889
Naive Bayes	0.57	0.53333	0.53333	0.66667	0.44444
Support Vector Machines	0.81	0.73333	0.75	0.85714	0.66667
K-Nearest Neighbor	0.57	0.6	0.625	0.71429	0.55556
Decision Trees	0.76	0.66667	0.61538	1	0.44444
Random Forest	0.65	0.6	0.625	0.71429	0.55556
Bootstrap-Aggregating Trees	0.68	0.6	0.625	0.71429	0.55556
Light GBM	0.81	0.73333	0.71429	1	0.55556
XG Boost	0.83	0.8	0.8	1	0.66667
Multilayer Perceptron	0.89	0.73333	0.75	0.85714	0.66667

## Data Availability

This manuscript has not been published and is not under consideration for publication elsewhere. The publication has been approved by all co-authors and the responsible authorities at the institute where the work has been carried out. The data presented in this study are available on request from the corresponding author.

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
