# Peer review of "Cervical Secretion Methylation Is Associated with the Pregnancy Outcome of Frozen-Thawed Embryo Transfer"

_ijms, 2023, doi:10.3390/ijms24021726_

Round 1

Reviewer 1 Report

This manuscript is devoted to the assessment of endometrial receptivity according to the methylome profile of HOXA10, HAND2, KSR1, PPT2, PRKAG2, and ZMIZ1 genes selected from the literature. The authors of the manuscript measured the methylation level of the six candidate genes with eight regions by the qMSP platform in the cervical secretions from 72 samples of women, 36 of whom became pregnant and 36 women did not. The machine learning approach showed that the combination of candidate genes’ DNA methylation profile could differentiate pregnancy from non-pregnancy samples as high as 86.67% accuracy with AUC 0.81.

I have a few questions and comments:

1) it is necessary to demonstrate the gene methylation status in the natural cycle and in the cycle with hormone replacement therapy on the day of embryo transfer; there might be differences

2) it is recommended to demonstrate the level of expression of proteins whose methylation status has been analyzed. Is there a relationship between protein expression levels and altered methylation status and pregnancy?

3) in the results and discussion it is necessary to provide information on the role of the protein products of the studied genes in the process of embryo implantation. The discussion is descriptive, with virtually no connection to the findings.

Reviewer 2 Report

The manuscript is clear, relevant for the field of interest, I could not find self citations.

I didn’t find any ethical concerns in the study and in data availability statements

Author Response

Many thanks for your comment.

Round 2

Reviewer 1 Report

Thank you so much for your responses to my comments. We look forward to continuing your research in this area.